# FluxHourly: Global long-term hourly 9 km terrestrial water-energy-carbon fluxes

Qianqian Han[1], Yijian Zeng[1], Yunfei Wang[1], Fakhereh (Sarah) Alidoost[2], Francesco Nattino[2], Yang Liu[2], Bob Su[1]

[1]Faculty of Geo-Information Science and Earth Observation (ITC), University of Twente, 7522 NH Enschede, the Netherlands
[2]Netherlands eScience Center, 1098 XH Amsterdam, the Netherlands

*Correspondence to*: Bob Su (z.su@utwente.nl)

**Abstract.** Land surface energy, water and carbon fluxes are key for understanding Earth's climate system, yet global continuous high resolution fluxes datasets remain scarce. In this study, we present a global long-term (2000-2020) hourly 9km dataset of terrestrial water-energy-carbon fluxes, generated by integrating model simulations, in-situ measurements, and machine learning with remote sensing and meteorological data. First, the integrated STEMMUS-SCOPE model was deployed to simulate land surface fluxes over 170 sites with in-situ measurements. The selected model-output variables include net radiation (Rn), latent heat flux (LE), sensible heat flux (H), soil heat flux (G), gross primary productivity (GPP), solar-induced fluorescence at 685 nm and 740 nm (SIF685, SIF740). Next, optimal interpolation was applied to merge Rn, LE, and H from STEMMUS-SCOPE simulations with eddy covariance observations. The optimal estimate of Rn, LE, H alongside STEMMUS-SCOPE simulated G, GPP, SIF685, SIF740 were then used as training data-pairs to develop the emulator using a multivariate Random Forest (RF) regression algorithm, referred to as Random Forest with Optimal Interpolation (RF_OI) to predict terrestrial water-energy-carbon fluxes. The results demonstrate that RF_OI can estimate land surface fluxes with Pearson Correlation Coefficient score (r-score) values higher than 0.88 except for GPP (Rn 0.99, LE 0.88, H 0.92, G 0.92, GPP 0.8, SIF685 0.99, SIF740 0.99). The testing results on independent stations (which were not included for developing the emulator) show r-score values higher than 0.8. The feature importance indicates that incoming shortwave radiation, surface soil moisture, and leaf area index are top predictor variables that determine the prediction performance. FluxHourly enables analysis of ecosystem responses to climate extremes at unprecedented spatiotemporal scales.

## 1 Introduction

Accurate estimation of the water-energy-carbon exchanges between terrestrial ecosystems and atmosphere plays a crucial role in understanding ecosystem functioning and climate interactions (Jung et al., 2020; Reich, 2010). The ecosystem-atmosphere water-energy fluxes, including net radiation (Rn), latent heat flux (LE), sensible heat flux (H), and soil heat flux (G), represent the transfer of energy between the land surface and the atmosphere. Rn captures the balance between incoming and outgoing radiation, providing insights into the availability of energy for ecosystem processes. LE quantifies

the energy exchange associated with evapotranspiration, reflecting the loss of energy through the conversion of liquid water into water vapor. H characterizes the transfer of heat due to temperature differences, influencing atmospheric stability and turbulent mixing processes. G refers to the diurnal transfer of heat flux into and from the subsurface (Gao et al., 2017).

Alongside these fluxes, carbon fluxes are crucial for understanding the carbon cycle dynamics within the land-atmosphere system. Gross primary productivity (GPP) is a key variable that represents the total amount of carbon dioxide assimilated by plants through photosynthesis. It plays a significant role in ecosystem carbon uptake and storage, affecting the atmospheric carbon dioxide concentrations and the overall carbon balance of ecosystems. Beyond GPP, another important indicator of photosynthetic activity is solar-induced chlorophyll fluorescence (SIF), which reflects the efficiency of plant photosynthesis

and carbon uptake (Sun et al., 2017). SIF provides a direct proxy for the light reactions of photosynthesis, enabling remote detection of vegetation physiological status and stress responses with higher sensitivity than traditional reflectance-based indices (Frankenberg et al., 2014; Guanter et al., 2014).

By incorporating these seven variables, we can gain a comprehensive understanding of the intricate interplay between water-energy-carbon fluxes. These variables provide insights into energy distribution, evapotranspiration, photosynthetic activity,

carbon uptake, water availability, and ecosystem functioning. Consistent datasets of these variables contribute to advancing our knowledge of ecosystem functioning and its response to climatic changes, which contribute further to sustainable land and water resources management practices and societal adaptation to climate changes (Zeng et al., 2025b).

In-situ measurements can be obtained with the eddy covariance method, which estimates the carbon, water, and energy fluxes (Baldocchi, 2014). The large-scale measurement network FLUXNET (Pastorello et al., 2020) and the harmonized

dataset PLUMBER2 (Abramowitz et al., 2024; Ukkola, 2020)—PLUMBER: Protocol for the Analysis of Land Surface Models (PALS) Land Surface Model Benchmarking Evaluation Project—integrate site-level flux measurements worldwide, providing detailed time series data (Baldocchi, 2008). However, eddy-covariance measurements are typically conducted at individual sites, usually representing areas of less than 1 km$^2$. Estimating these fluxes at regional to global scales requires spatial upscaling to account for the heterogeneity and variability of these processes across larger areas. Previous efforts to

integrate FLUXNET measurements with satellite remote sensing, reanalysis data, and machine learning techniques have yielded global estimates of land-atmosphere fluxes. An representative example is the FLUXCOM product with a spatial-temporal resolution of 0.0833° & 8-daily and 0.5° & daily. These estimates have been extensively used to evaluate land surface models, assess water budgets, and investigate land-atmosphere interactions (Jung et al., 2019).

The flux products mentioned above were estimated at coarse temporal resolutions, limiting their ability to capture diurnal

variability and finer-scale processes (Jung et al., 2019; Jung et al., 2020; Tramontana et al., 2016). However, understanding the dynamics of water-energy-carbon fluxes at finer temporal scales is crucial for capturing rapid changes, short-term fluctuations, and interactions within the land-atmosphere system. High-frequency flux data allow for a more detailed investigation of diurnal patterns, environmental responses, and the interplay between these fluxes, ultimately improving our understanding of ecosystem processes and its responses to extreme climates. Therefore, there is a pressing need to move

beyond coarse temporal resolutions and obtain flux estimates at higher temporal resolutions - such as hourly intervals

comparable to in-situ observations - to accurately capture the temporal dynamics and fully unravel the complexities of the land-atmosphere system. To address this need, at each PLUMBER2 site (Section 2), we applied the soil-plant model— STEMMUS-SCOPE (Wang et al., 2021; Zeng et al., 2025a) to retrieve hourly time series of all seven target variables (Rn, LE, H, G, GPP, SIF at 685 nm and SIF at 740 nm) (Section 3). We then combined these model outputs with the corresponding tower observations to train a random forest upscaling algorithm, which generates a global hourly 0.1° (9 km) fluxes from 2000 to 2020 (FluxHourly) (Section 4). Furthermore, we discuss the potential limitations and corresponding uncertainties of FluxHourly datasets (Section 5), and draw our conclusion accordingly (Section 6).

## 2 Data

### 2.1 Flux data for training and testing

PLUMBER2 presented a dataset that includes 170 globally distributed flux tower sites (Figure 1), from FLUXNET2015, and complemented from La Thuile, and OzFlux networks (Ukkola et al., 2022). The earliest available in-situ measurements are from 1992 and a majority of the sites have data available until 2014 (some sites until 2018). The duration of observations for each tower site was illustrated in our previous work (Wang et al., 2025). In total, the PLUMBER2 dataset includes 170 sites spanning from 1992 to 2018, corresponding to over 1000 site-years of observations. The 170 sites cover 17 Köppen climate zones, with the following distribution: Af (2), Am (3), Aw (6), BSh (8), BSk (10), BWk (1), Cfa (22), Cfb (42), Csa (20), Csb (6), Dfa (5), Dfb (13), Dfc (23), Dwa (1), Dwb (2), DWc (1), and ET (5).

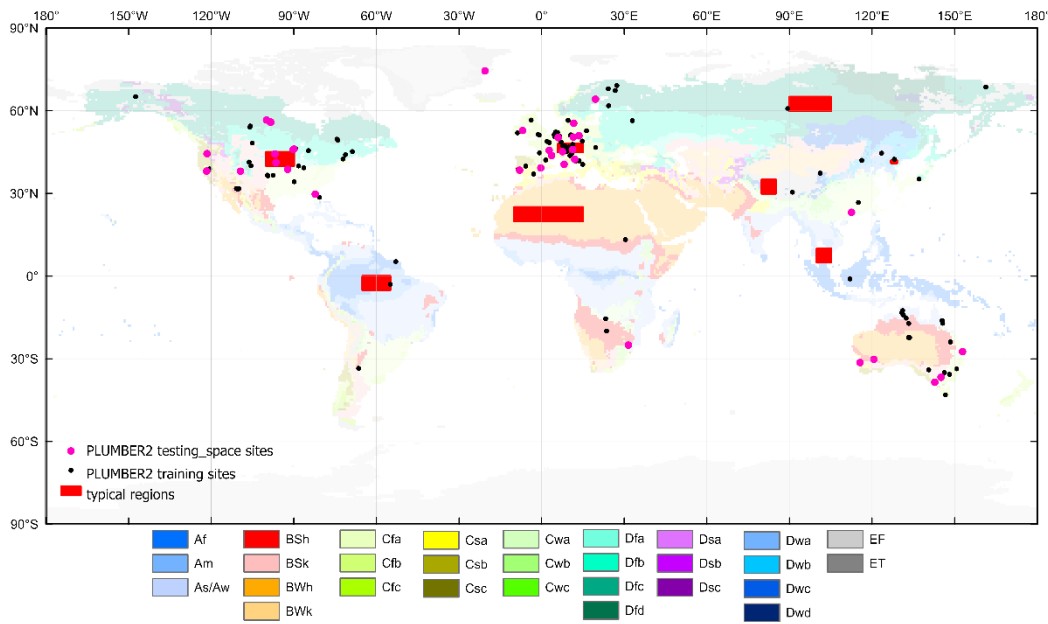

Figure 1. Spatial distribution of PLUMBER2 sites considered in this study and their corresponding climate zones, as well as the eight typical regions selected (based on Köppen–Geiger (KG) climate classification system) for seasonal comparisons and showing diurnal characteristics of FluxHourly data.

The training and testing datasets for Random Forest (RF) algorithm were derived from STEMMUS-SCOPE input and output data, and PLUMBER2 in-situ measurements. These datasets comprise 170 observation stations covering 11 distinct IGBP land cover types, and have half-hourly temporal resolution. The input variables and target variables are listed in Table 1.

To account for uncertainty information from both the STEMMUS-SCOPE model and PLUMBER2 datasets, optimal interpolation was applied to merge Rn, LE, and H from STEMMUS-SCOPE simulations with eddy covariance observations. It is to note that although PLUMBER2 provides Rn, LE, H, GPP, and G, the metadata on the installation depth for sensors measuring G was not provided. As such, only Rn, LE, and H were used for optimal interpolation. The optimal interpolated Rn, LE, H alongside STEMMUS-SCOPE simulated G, GPP, SIF685, SIF740 were used as target variables in the training and testing datasets. Prior to the training, the energy-closure based data quality control was implemented (see Supplementary S1 for the data preprocessing protocol).

Data splitting was performed based on spatial, temporal, and random criteria. For each IGBP type, 20% of the data was reserved for testing in space (34 stations covering 11 IGBP), while the remaining 80% was further split into an 80% training and random testing subset and a 20% testing subset based on time. Finally the data was aggregated to hourly from half-hourly to reduce data volume.

## 2.2 Satellite and meteorological data

We collected seven aerodynamic predictor variables from European ReAnalysis Land (ERA5-Land, Table 1). ERA5-Land describes the evolution of water and energy cycles over land with 69 variables available from 1981 until now in a consistent manner, which, among others, could be used to analyse trends and anomalies (Muñoz-Sabater et al., 2021). The spatial and temporal resolution used in this study is 9 km and hourly. We also collected three canopy related predictor variables and $CO_2$ concentration and surface soil moisture. The IGBP land cover type is converted from land cover.

Table1  Input and output variables.

| abbreviations | Full name | unit | Dataset source |
|---|---|---|---|
| Input variables | | | |
| ea | Air vapour pressure | hPa | ERA5-Land |
| p | air pressure | hPa | ERA5-Land |

| Rli | Incoming longwave radiation | W/m$^2$ | ERA5-Land |
|---|---|---|---|
| Rin | Incoming shortwave radiation | W/m$^2$ | ERA5-Land |
| Ta | Air temperature | degree | ERA5-Land |
| u | Wind speed | m/s | ERA5-Land |
| Precip | Precipitation | mm | ERA5-Land |
| hc | Canopy height | m | (Lang et al., 2023) |
| LAI | Leaf area index | m$^2$/m$^2$ | https://land.copernicus.eu/en/products/vegetation/leaf-area-index-v2-0-1km |
| Vcmax | Maximum carboxylation rate | µmol/m$^2$/s | (Chen et al., 2022) |
| CO2 | Carbon dioxide concentration | mg/m³ | https://ads.atmosphere.copernicus.eu/cdsapp#!/dataset/cams-global-ghg-reanalysis-egg4?tab=overview |
| SSM | Surface soil moisture | m$^3$/m$^3$ | (Han et al., 2023) |
| IGBP | International Geosphere–Biosphere Programme | | https://cds.climate.copernicus.eu/cdsapp#!/dataset/satellite-land-cover?tab=overview |
| Output variables | | | |
| Rn | Net radiation | W/m$^2$ | |
| LE | Latent flux | W/m$^2$ | |
| H | Sensible flux | W/m$^2$ | |
| G | Soil heat flux | W/m$^2$ | |
| GPP | Gross primary productivity | µmol/m$^2$/s | |
| SIF685 | Solar-induced chlorophyll fluorescence (SIF) at 685 nm | W/m$^2$/um/sr | |
| SIF740 | Solar-induced chlorophyll fluorescence (SIF) at 740 nm | W/m$^2$/um/sr | |

We used three existing flux datasets and one satellite SIF product for comparison, including FLUXCOM (Jung et al., 2019), FLUXFORMER (Phan and Fukui, 2024), GLEAM (v4.2a) (Miralles et al., 2024), and TROPOMISIF (Guanter et al., 2021).

## 3 Methods

### 3.1 Compute Platform

We conducted the whole intensive computing at the Dutch National Supercomputer 'Snellius' (https://www.surf.nl/en/dutch-national-supercomputer-snellius), using Dask for parallel computing. The used input data volume is around 1.5 TB per year, and output data volume around 0.5 TB per year, with the zarr data format. For producing one month data of FluxHourly, it took 7 minutes with the setting of 128 cores and 960 GB memory. Figure 2 presents the whole workflow.

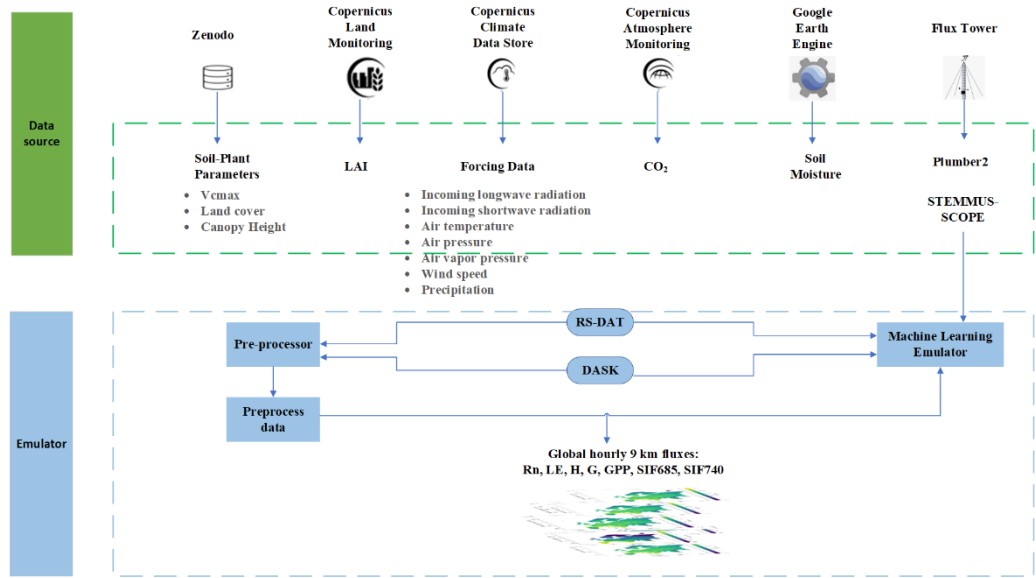

Figure 2. Flowchart of producing FluxHourly.

RS-DAT: Remote Sensing Deployable Analysis environment (https://research-software-directory.org/projects/rs-dat)

### 3.2 STEMMUS-SCOPE

The STEMMUS-SCOPE (STEMMUS- Simultaneous Transfer of Energy, Mass, and Momentum in Unsaturated Soil; SCOPE - Soil Canopy Observation, Photochemistry and Energy fluxes radiative transfer) integrates radiative transfer, energy balance at leaf and canopy scale with the soil water and soil heat dynamics and root development to simulate the exchange of water, energy, and carbon between the atmosphere, vegetation, and soil (Wang et al., 2021). It calculates the profiles of soil moisture and soil temperature, top-of-canopy outgoing radiation, net radiation components (shortwave, longwave), solar-induced fluorescence (SIF), and absorbed photosynthetically active radiation (aPAR) as well as turbulent water and carbon fluxes (Tol et al., 2009; Yu et al., 2018; Zeng et al., 2011a, 2011b). Additionally, it represents coupled liquid, vapor, dry air, and heat transfer in multi-layer unsaturated soil (Zeng et al., 2011a, 2011b). By coordinating carbon assimilation and allocation with soil water and heat dynamics, STEMMUS-SCOPE provides detailed insights into canopy radiation, soil

processes, and land-atmosphere fluxes, and can be used to produce eco-hydrological datasets across diverse vegetation types (Wang et al., 2021; Zeng et al., 2025a).

### 3.3 Machine Learning Method

Random Forest (RF) algorithm is an ensemble learning method that outputs a result based on the mean of the many individual training models (trees) (Breiman, 2001). RF follows the Bootstrap Aggregation (Bagging) strategies, i.e. random sampling with replacement (Altman and Krzywinski, 2017). The RF algorithm was trained to learn the relationship between
the 13 input predictor variables and 7 target variables from STEMMUS-SCOPE output (Table 1) and PLUMBER2 measurements to develop an emulator (Zeng et al., 2025a).

### 3.4 Optimal interpolation

Optimal interpolation, a simple data assimilation method was employed to merge in-situ measurements and model outputs
(Oke et al., 2010). This aims to take advantage of consistency of model physics that serves as background (or a prior) and in-situ observation as independent measurement. As such in-consistencies in in-situ observation due to malfunction of instruments and gap filling of missing data can be detected and removed. For in-situ data and model output for Rn, LE, H, we used optimal interpolation to merge each variable separately, while for G, SIF685, SIF740, and GPP, we used only model output, since in-situ measurements are not available.

Optimal interpolation assumes that both model outputs and observations contain errors characterized by their error variances and covariances. The method provides the best linear unbiased estimate (BLUE) of the target variables by minimizing the expected mean-square error, which is mathematically equivalent to a weighted least squares solution::

$$x = w1 * x1 + w2 * x2 \quad (1)$$

where: $w1 = \frac{var2}{var1+var2}$, $w2 = \frac{var1}{var1+var2}$, var1 = variance(x1), var2 = variance(x2), x1 is in-situ data, x2 is model output.

In this formulation, daily variances for each data source were calculated based on 48 half-hourly values. The idea is to assign greater weight to the source that shows less fluctuation within a day, reflecting more stable data quality. This approach is especially beneficial in handling periods where eddy covariance observations may be less reliable. For instance, in situations such as nighttime periods or rainfall events where eddy covariance data tend to be noisy or unreliable, the STEMMUS-
SCOPE simulations which is physically constrained receives more weight.

**3.5 Evaluation metrics**

To assess the performances of RF, we compared predicted fluxes with in-situ measurements, using two commonly used statistical evaluation metrics (Entekhabi et al., 2010): the Root Mean Square Error (RMSE) (Eq. 2), and the Pearson Correlation Coefficient (r) score (Eq. 3), as follows.

$$RMSE = \sqrt{\frac{\sum_{i=1}^{N}(y_{pred,i} - y_{ref,i})^2}{N}} \tag{2}$$

$$r = \frac{\sum_{i=1}^{N}(y_{pred,i} - \overline{y_{pred,i}})(y_{ref,i} - \overline{y_{ref,i}})}{\sqrt{\sum_{i=1}^{N}(y_{pred,i} - \overline{y_{pred,i}})^2}\sqrt{\sum_{i=1}^{N}(y_{ref,i} - \overline{y_{ref,i}})^2}} \tag{3}$$

In Eqs. 2 and 3, $y_{pred,i}$ is the predicted fluxes, $y_{ref,i}$ is in-situ measured fluxes, N is the number of valid pairs of fluxes, $\overline{y_{pred,i}}$ is the mean value of the predicted fluxes, $\overline{y_{ref,i}}$ is the mean value of in-situ measured fluxes.

**4 Results**

**4.1 Testing on site level**

**4.1.1 General performance on three testing sets**

The emulator RF_OI was developed by training RF with input data from PLUMBER2, output data from optimal interpolated result from STEMMUS-SCOPE simulations and PLUMBER2 measurements. The performance on site level was tested on three testing datasets for the emulator: testing_random (for testing the performance of randomly split data during training), testing_time (for testing the performance on the period that was not used in training), testing_space (for testing the performance on the stations that were not used in training).

Table 2 demonstrates the performance of the RF_OI emulator across three testing sets. The results show consistently high accuracy for Rn, SIF685 and SIF740, with RMSE values below 36.03 W m⁻² (for Rn) and 0.04 W m⁻² μm⁻¹ sr⁻¹ (for SIF685 and SIF740), and r scores reaching 0.99 across all sets. LE, H, G also exhibit good performance, though their RMSE and r score values indicate slightly lower accuracy, particularly for testing_space set. GPP shows moderate performance with RMSE around 4.30–4.86 μmol m⁻² s⁻¹ and r scores near 0.80. Overall, RF_OI achieves reliable predictions across variables, with the best performance observed for Rn, SIF685 and SIF740.

Table 2 Performance metrics obtained by the emulator RF_OI on three testing sets

| Target variables | Testing_random | | Testing_time | | Testing_space | |
|---|---|---|---|---|---|---|
| | RMSE | r score | RMSE | r score | RMSE | r score |
| Rn (W m$^{-2}$) | 31.85 | 0.99 | 31.20 | 0.99 | 36.03 | 0.99 |
| LE (W m$^{-2}$) | 37.44 | 0.88 | 40.56 | 0.87 | 39.76 | 0.80 |
| H (W m$^{-2}$) | 38.01 | 0.92 | 39.42 | 0.91 | 51.40 | 0.89 |
| G (W m$^{-2}$) | 15.85 | 0.92 | 17.05 | 0.90 | 18.30 | 0.86 |
| GPP (umol m$^{-2}$ s$^{-1}$) | 4.77 | 0.80 | 4.86 | 0.79 | 4.30 | 0.81 |
| SIF685 (W m$^{-2}$ um$^{-1}$ sr$^{-1}$) | 0.01 | 0.99 | 0.01 | 0.99 | 0.01 | 0.99 |
| SIF740 (W m$^{-2}$ um$^{-1}$ sr$^{-1}$) | 0.02 | 0.99 | 0.03 | 0.99 | 0.04 | 0.99 |

Figure 3 presents key predictor variables in the RF_OI emulator, using permutation importance. Across all seven target variables, the most important feature is Rin. Rin provides the primary energy input for surface heating, evapotranspiration, and photosynthesis, thereby exerting strong control over both energy and carbon fluxes. For the energy components (Rn, H, LE, G), Rin determines the available energy that can be partitioned into different fluxes (Peng et al., 2021). For the carbon-related processes (GPP and SIF), Rin governs photosynthetically active radiation (PAR), which supplies the energy for carbon assimilation (Nemani et al., 2003; Running et al., 2004). These findings are therefore physically consistent with the observed dominance of Rin in the feature importance analysis. For net radiation Rn, Rin, Rli, and Ta are the most important variables, underscoring the essential role of radiation and temperature in driving net radiation. For latent heat flux LE, Rin is ranked as the most important variable with subsequent importance variables SSM and Ta. From a physical process perspective, both SSM and LAI are closely related to latent heat flux, which suggests that both methods account for the physical consistency of soil-plant processes that influence LE. For sensible heat flux H, the top three variables are Rin, SSM and Ta. In the case of G, Rin is the most important variable, and u and Ta are the second and third important features. Physically, besides Rin, Ta and u significantly influence ground heat flux, as they directly affect the temperature gradient between the surface and the atmosphere, driving heat transfer from the ground. For gross primary productivity GPP, as well as SIF685 and SIF740, the top three predictor variables, namely Rin, LAI, and SSM, reinforces the importance of radiation, vegetation structure, and soil moisture in these processes.

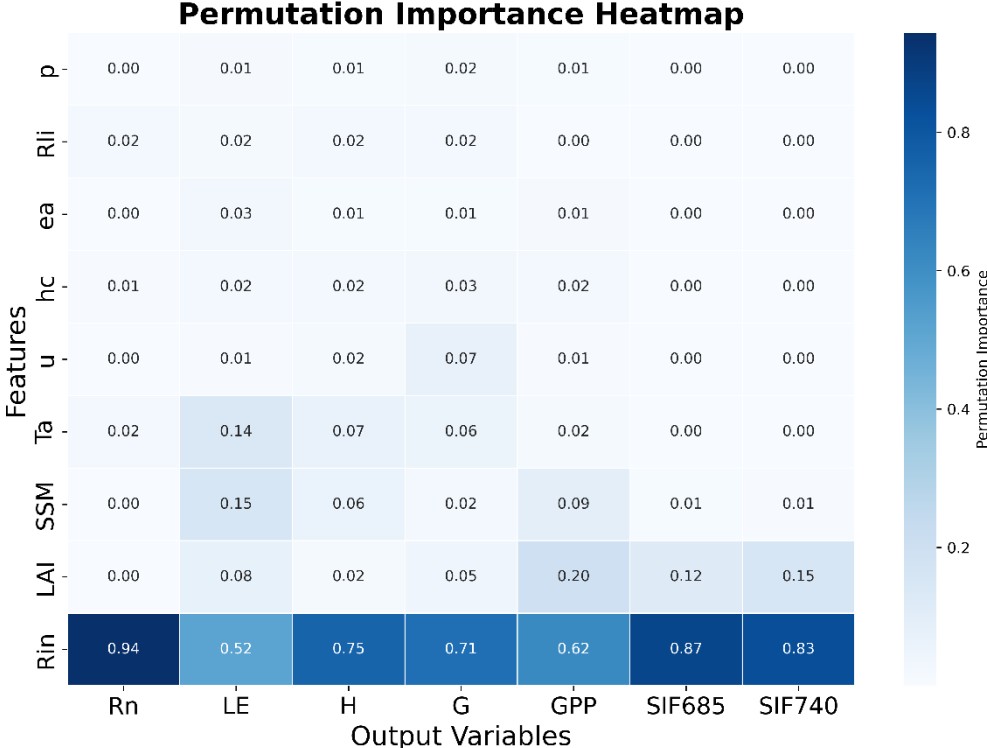

Figure. 3 permutation feature importance of RF_OI on Rn, LE, H, G, GPP, SIF685, SIF740.

**4.1.2 Dynamic features**

We analyzed the probability density functions (PDFs) of five datasets—in-situ fluxes, FluxHourly, FLUXCOM, GLEAM, FLUXFORMER—across 34 stations within the testing_space set over the period from 2001 to 2014 (Figure 4). The PDFs were computed to assess the distribution characteristics of daily Rn, LE, H and monthly GPP, keeping the same temporal resolution for all these datasets. By evaluating the intersection ranges between the PDFs of these datasets and in situ fluxes, we assessed their consistency and performance.

The overlapping ranges between in-situ measurements and various datasets revealed significant insights into their consistency. For Rn, FluxHourly demonstrated a higher degree of agreement with in-situ Rn, covering 95.3% of the shared distribution, compared to an 84.4% overlaping area for FLUXCOM. This indicates that FluxHourly aligns more closely with in-situ Rn measurements than FLUXCOM. For LE, FLUXCOM showed a slightly higher degree of agreement with in-situ LE at 86.8%, while FluxHourly followed closely with 86.7%. GLEAM also performed well, with an 85.6% agreement. In the case of H, GLEAM exhibited the highest agreement with in-situ H at 87.6%, slightly outperforming FluxHourly (86.9%) and significantly surpassing FLUXCOM (81.7%). For GPP, FLUXFORMER achieved the highest agreement with in-situ GPP at 88.4%, while FluxHourly and FLUXCOM showed slightly lower agreement of 86.6% and 85.7%, respectively.

FLUXCOM and FLUXFORMER overestimated GPP by 6-9 gC/m²/day and underestimated it between 0 and 2 gC/m²/day while FluxHourly overestimated around 5-12 gC/m²/day.

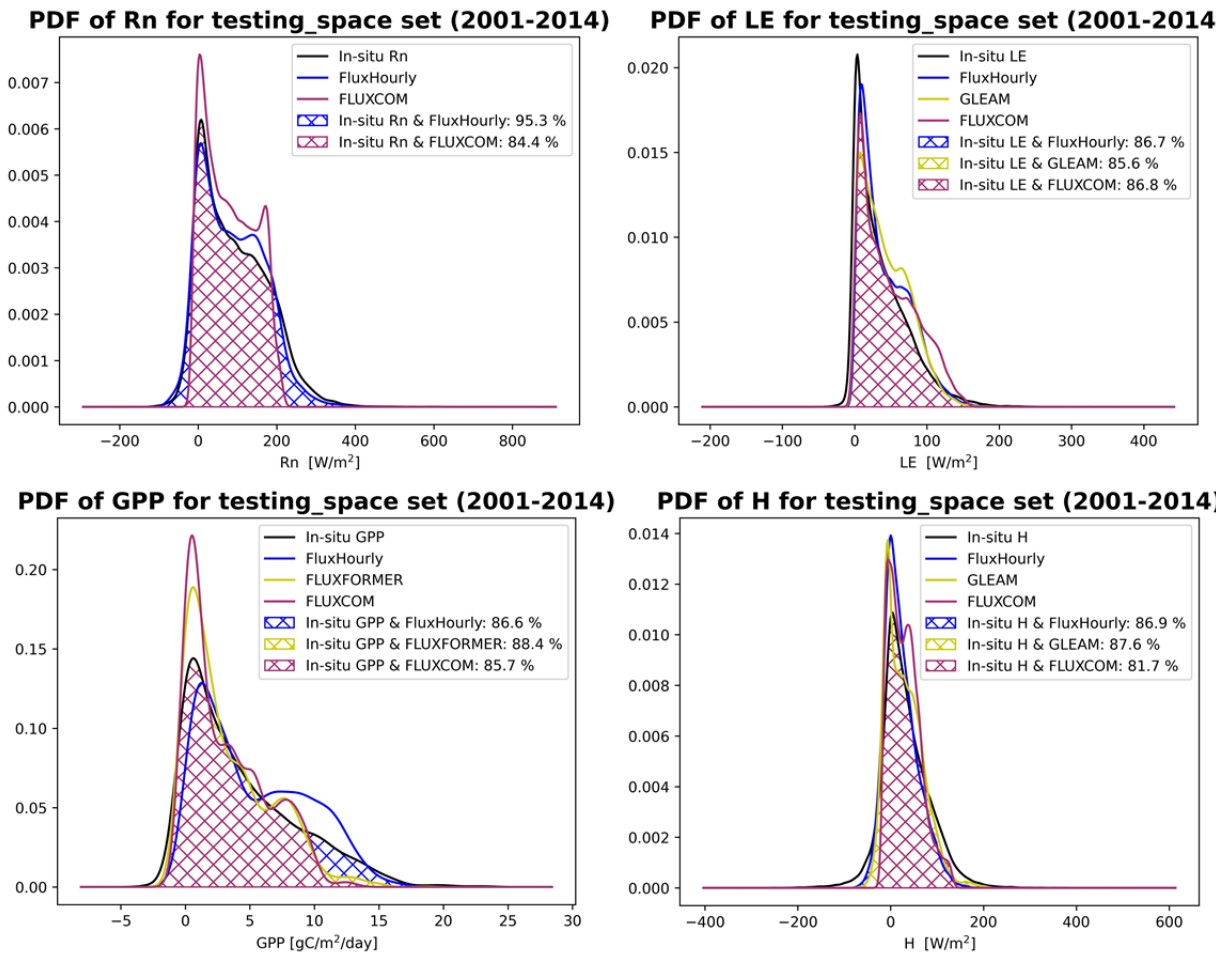

Figure 4. PDF of Rn, LE, H, GPP for independent sites (testing_space set) (2001-2014) among in-situ fluxes, FluxHourly, GLEAM, FLUXCOM, FLUXFORMER. Note: shaded area indicates the percentage of agreement between a data product
and in-situ data.

While the analysis of overlapping ranges across 34 testing_space stations provides a comprehensive overview of dataset performance, it is also valuable to compare the temporal dynamics of fluxes at individual sites to gain deeper insights. We selected three representative sites for detailed analysis in order to further reveal the temporal variation characteristics of fluxes of different ecosystems. These sites include evergreen broadleaf forests (IT-Lav), mixed forests (BE-Vie), and
permanent wetlands (US-Los), which represent contrasting vegetation structures, hydrological conditions, and energy-water exchange regimes. Together, they cover key ecosystem types from high-productivity forests to high-water wetlands, providing complementary conditions for testing the robustness of the model. By comparing the time series at these three sites on monthly (Figure S6) and their original temporal resolution of each products (Figure 5, also see Figure S7-9), we can have

a deeper understanding of the impact of ecosystem types and environmental conditions on flux dynamics. To present the data

more clearly, we calculated multiple years average and standard deviation (std) on monthly scale (Figure S6). For multiple

scales comparison, we show several days of data (Figure 5). For hourly comparison, we show several days of data in summer

and winter respectively (Figure S7-9).

Figure S6 compares monthly multiple years average and std in Rn, LE, H, and GPP across three stations (BE-Vie, IT-Lav, and US-Los).

To highlight the differences in temporal resolution among datasets, we compared the raw time-series data from FluxHourly,

FLUXCOM, GLEAM, and FLUXFORMER at station BE-Vie (Figure 5). From 12 May to 19 May, FLUXCOM provides 8

values for Rn, LE, and H, while GLEAM provides 8 values for LE and H. In addition, FLUXCOM and FLUXFORMER

provide only one value for GPP during the same period. FluxHourly, however, captures diurnal variations and demonstrates

strong consistency with in-situ measurements, highlighting its ability to capture fine-scale temporal dynamics of carbon

fluxes.

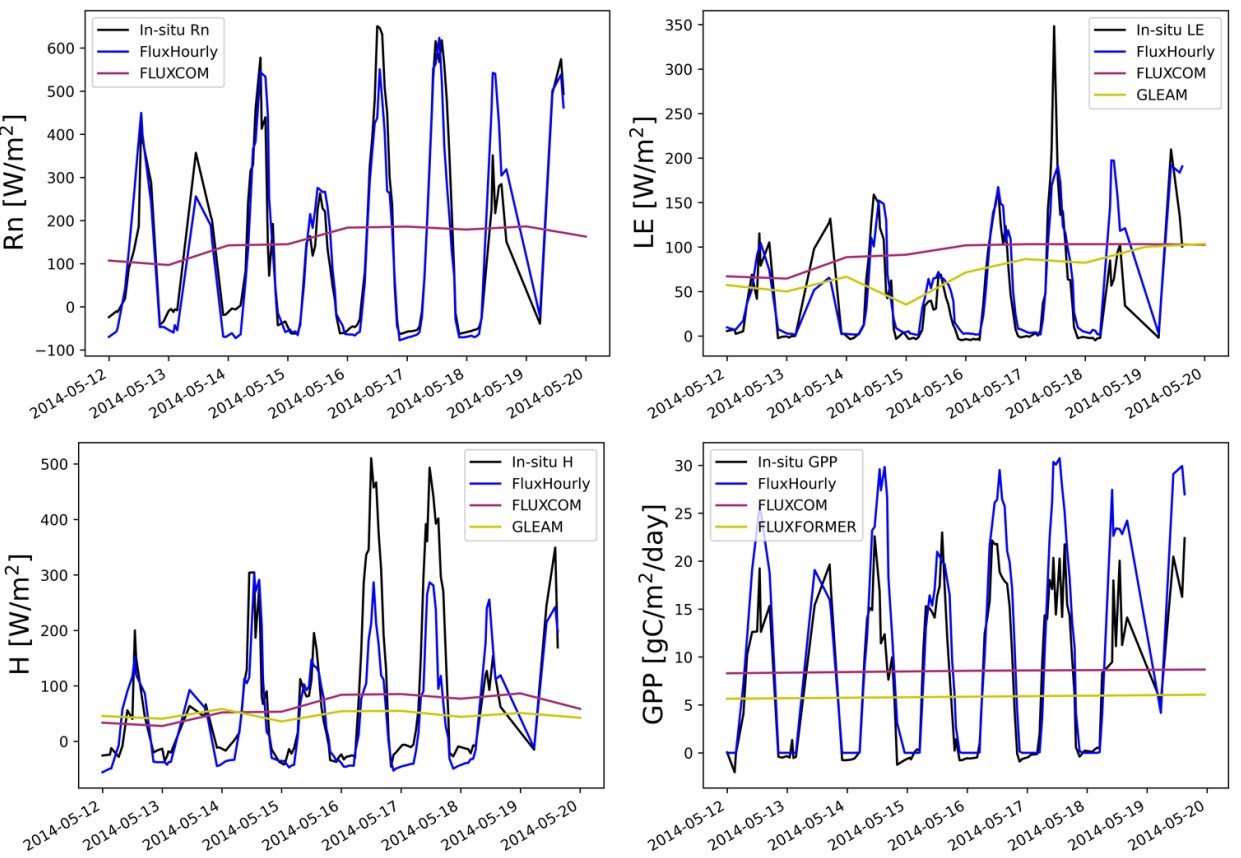

Figure 5. Hourly FluxHourly, daily/monthly FLUXCOM and FLUXFORMER, daily GLEAM at station BE-Vie: Rn, LE, H,

GPP  (Time is local time)

**4.2 Global flux products with hourly resolution**

**4.2.1 Inter comparison with existing flux datasets on spatial pattern**

FluxHourly is a global hourly 9 km flux dataset between 2000-2020, but FLUXCOM has data only until 2014. As such, we calculated annual mean of FLUXCOM and FluxHourly in 2014 for fair comparison. For SIF, as TROPOMISIF has data from May 2018 until April 2021 (Guanter et al., 2021), we conducted the comparison in May 2018. Figures 6 and S9 show the spatial pattern and the latitudinal profile.

Figures 6 and S9 illustrates the spatial distribution of Rn, LE, H, GPP and SIF740 across different datasets. High-value regions for Rn and LE are primarily in the tropics, reflecting the abundance of solar energy and active energy exchanges in these areas. In contrast, low-value regions are observed in high-latitude areas. Sensible heat also exhibits notable spatial distribution, featuring two peaks: a smaller peak around 15°N latitude and a larger peak around 25°S latitude. This distribution aligns with the climatic characteristics of tropical and high-latitude regions, highlighting geographical differences in Rn, LE, and H within the global climate system. The latitude profiles further display the differences among the datasets by each latitude degree, indicating that Rn, LE, and H values are significantly higher in the tropics and considerably lower in high latitudes. While FluxHourly and FLUXCOM exhibit similar trends in both Rn and LE, FluxHourly shows a lower magnitude of LE compared to the FLUXCOM and GLEAM in the tropics. Additionally, FluxHourly and GLEAM demonstrate similar trends for LE between 30°S and 30°N, while the FLUXCOM reveals discrepancies with the other two datasets at certain latitudes. While the three datasets exhibit consistent latitude dynamics of H, between the equator and 35°S, GLEAM demonstrates the highest values for H, with FLUXCOM in the middle and FluxHourly showing the lowest values across the datasets.

In Figure S10, global GPP spatial maps illustrate the distribution of GPP across different datasets. High-value regions are primarily concentrated in the tropics, indicating abundant photosynthetic activity in these areas. In contrast, significant low-value regions are observed in high-latitude areas, where GPP approaches zero. The latitude profiles indicate that GPP values are higher in the tropics, with four notable peaks around 50°N, 20°N, 10°S, and 45°S; however, between 30°N and 60°N, both FLUXCOM and FLUXFORMER exhibit lower values.

Figure S10 illustrates also the distribution of SIF across two datasets. High-value regions are primarily located around 45°N and 0° to 25°S, indicating abundant photosynthetic activity in these areas. In contrast, significant low-value regions are observed in the Sahara Desert, central Asia, the Arctic, and Oceania. The latitude profiles show that FluxHourly and TROPOMI SIF exhibit similar patterns and trends; however, FluxHourly consistently records higher SIF values compared to TROPOMISIF.

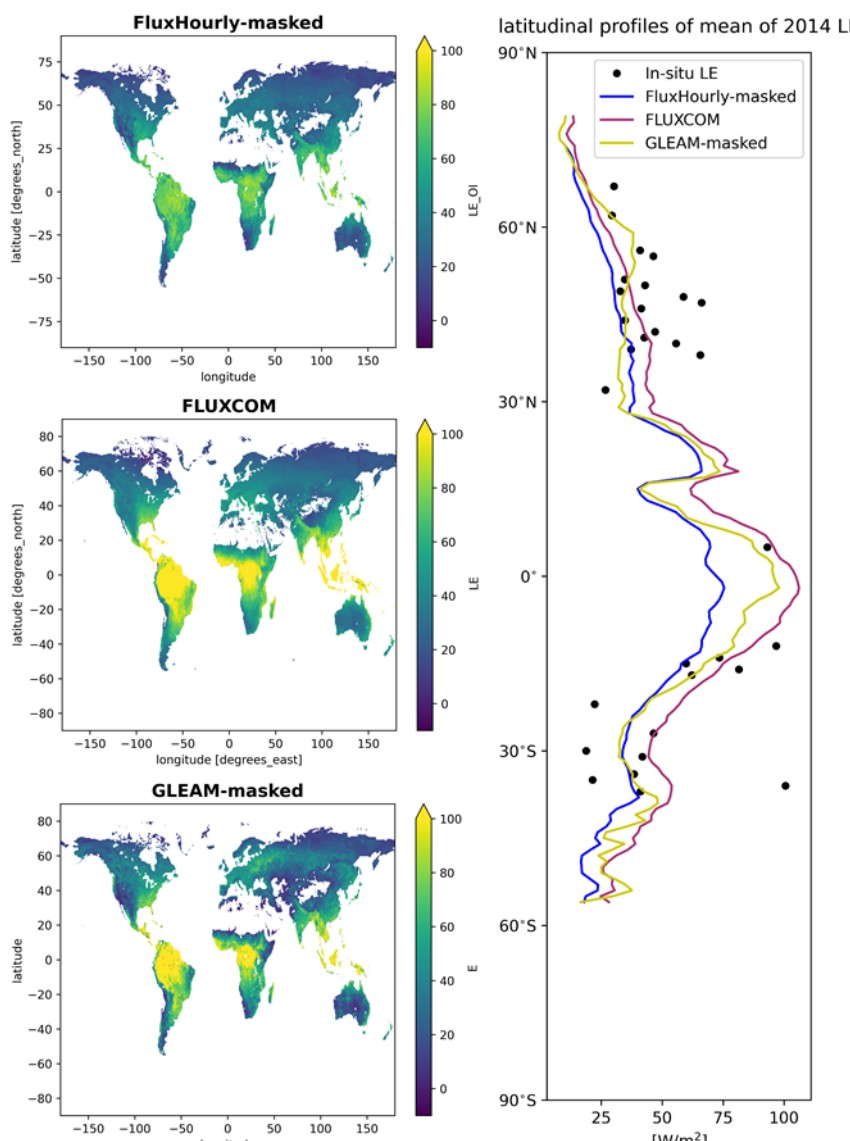

Figure 6. Annual mean of predicted hourly 9 km LE by RF_OI, and FLUXCOM, GLEAM in 2014 (FLUXCOM is used as a mask for FluxHourly and GLEAM because it has missing data)

**4.2.2 Diurnal and seasonal changes**

We investigated FluxHourly's ability to capture diurnal variations of fluxes across the globe in 2014, since the sub-daily resolution is an important feature of the new dataset. We selected eight typical regions based on Köppen–Geiger (KG) climate classification system (Figure 1). The KG system classifies the climate based on air temperature and precipitation.

The climate is grouped into 5 main classes with 30 sub-types, consisting of tropical, arid, temperate, continental, and polar climates (Beck et al., 2018). The eight selected regions cover five main climate zones (Table 3).

Table 3 Detail information of 8 typical regions

| Regions | Longitude | Latitude | Time zone | Köppen–Geiger climate |
|---|---|---|---|---|
| Amazon Rainforest | -65°W - -55°W | -5°S - 0°N | UTC-3 | Tropical: Af, Am |
| Southeast Asia Mangroves | 100°E - 105°E | 5°N - 10°N | UTC+8 | Tropical: Am, Af |
| Sahara Desert | 10°W - 15°E | 20°N - 25°N | UTC+1 | Arid: Bwh |
| Alps | 6°E - 15°E | 45°N - 48°N | UTC+1 | Temperate: Cfa,Cfb, Continental: Dfb, Dfc, Polar: ET |
| Siberian Tundra | 90°E - 105°E | 60°N - 65°N | UTC+8 | Continental: Dfc |
| Himalayas | 80°E - 85°E | 30°N - 35°N | UTC+5 | Polar: ET Continental: Dwc |
| North American Grasslands | -100°W - -90°W | 40°N - 45°N | UTC-6 | Continental: Dfa, Dfb |
| Changbai Mountain Forest | 127°E - 129°E | 41°N - 42°N | UTC+8 | Continental: Dwb, Dwc |

As expected, the ensemble diurnal cycles are clearly shown in FluxHourly at all our eight selected locations of the globe (Figures 7-8) for LE and GPP, while those for Rn, H, G are given in supplementary Figures S10-S12.

In humid regions like tropical rainforests, Rn and LE remain high and stable throughout the year, driven by abundant solar radiation and water availability. The diurnal variation is similar for all seasons due to continuous evapotranspiration. In contrast, arid regions, such as the Sahara Desert, exhibit strong seasonal and diurnal fluctuations. Cold regions, including Siberia and the Himalayas, have low Rn and LE in winter, with brief summer peaks during the growing season. Temperate regions, like grasslands, show clear seasonal patterns, with higher values in spring and summer and lower values in winter. H and G follow similar trends: in humid regions, H remains low due to dominant evapotranspiration, while in arid regions, H

increases significantly during the day as heat is transferred from the surface to the atmosphere. Cold regions see a rise in H during winter due to reduced evapotranspiration, whereas in temperate regions, H varies seasonally. G remains small and stable in humid regions but fluctuates significantly in arid, cold, and temperate regions, with pronounced diurnal variations in summer when soil heat transfer is more active.

In humid regions, GPP is consistently high due to favorable climate conditions, with peak values during daylight hours. In arid regions, it is highly seasonal, occurring mainly during wet periods, while in cold regions, GPP is restricted to the short growing season, peaking in summer and nearing zero in winter. Temperate regions show strong seasonal variations, with higher GPP in spring and summer and reduced photosynthetic activity in colder months. The seasonal change in diurnal cycle of GPP is minimal in humid regions but more pronounced in arid, cold, and temperate regions.

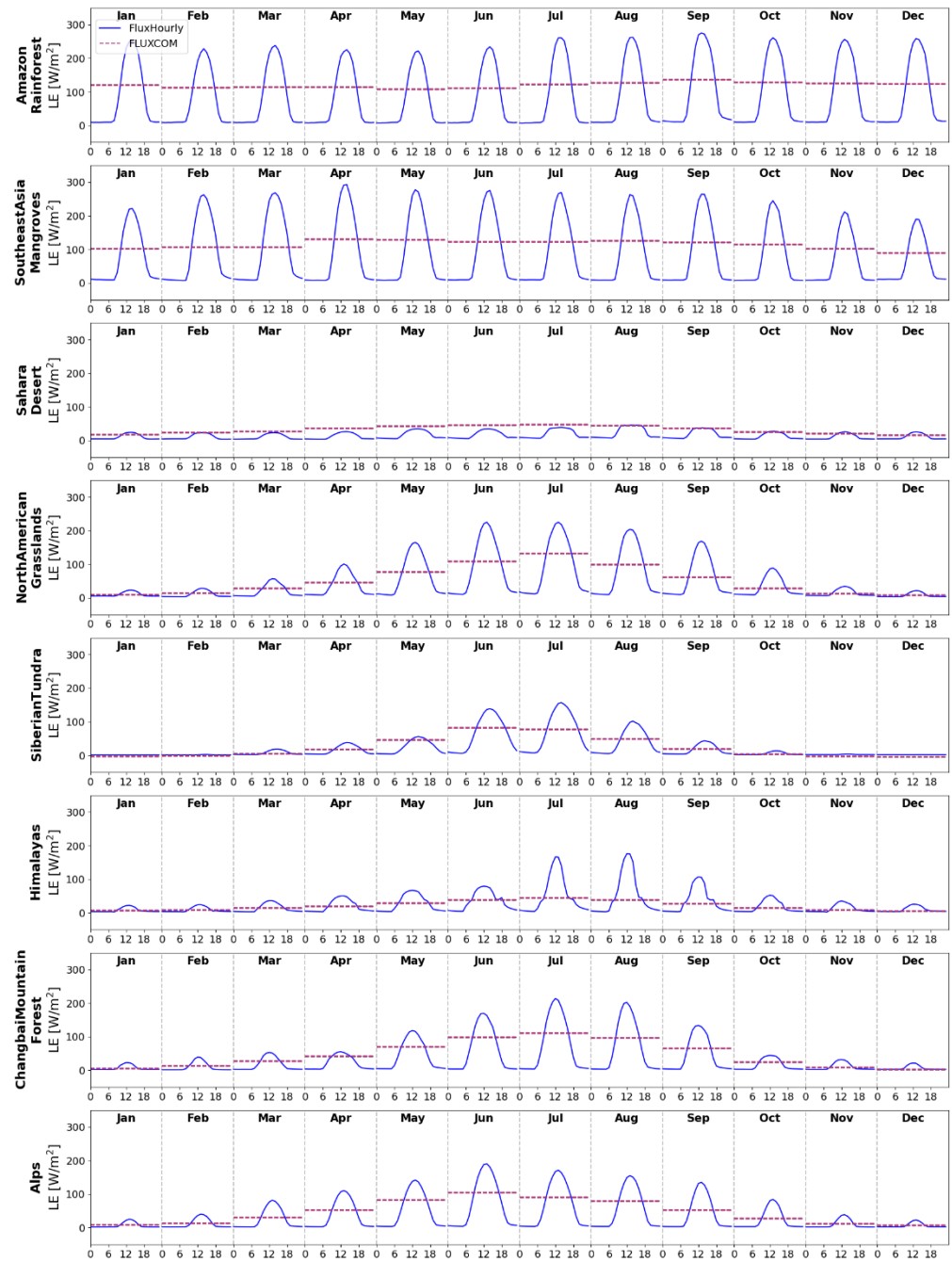

Figure 7. Diurnal cycles of LE for 8 regions for each month of the year, where each panel refers to a region. The data is converted to local time zone.

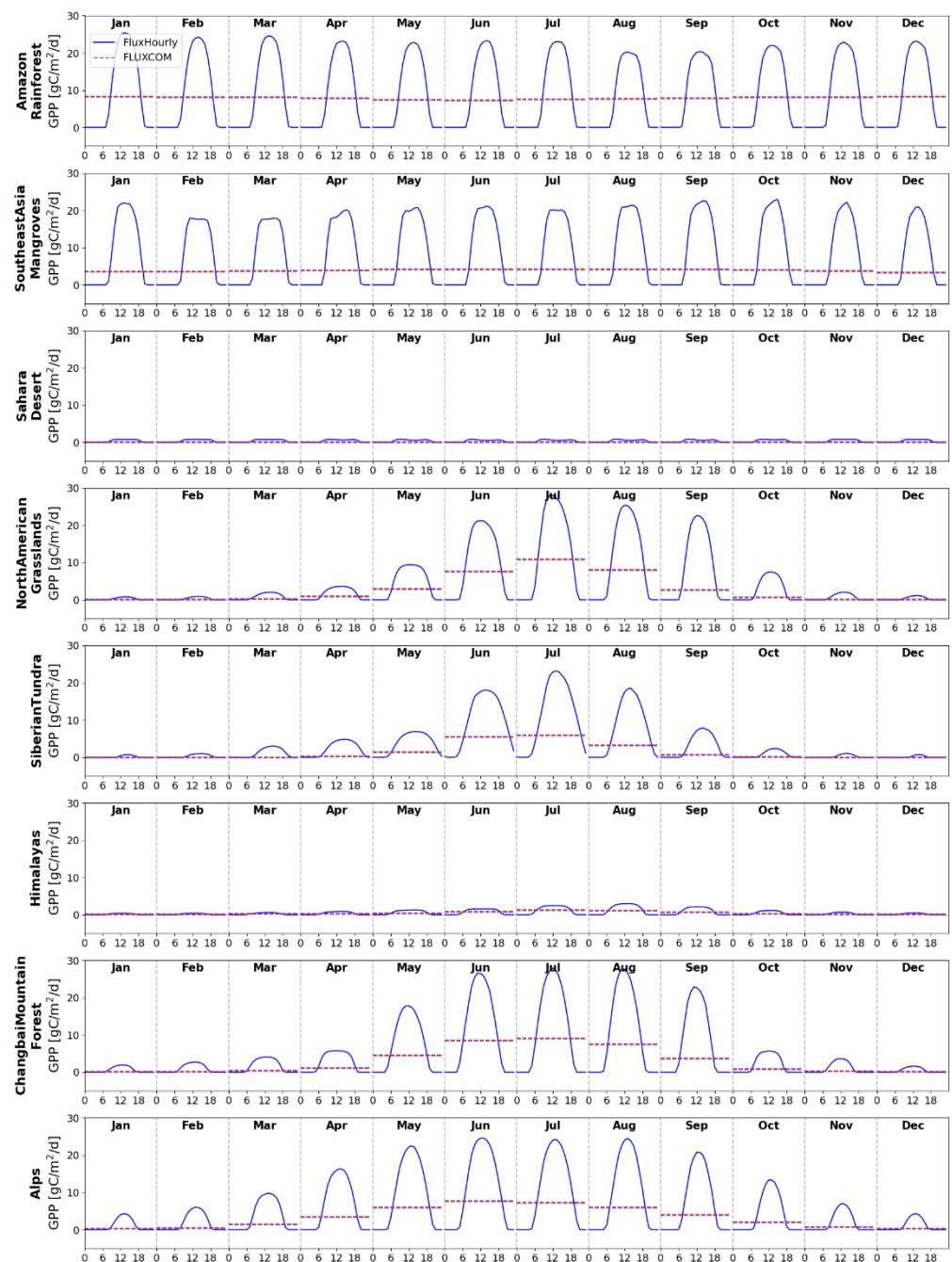

Figure 8. Diurnal cycles of GPP for 8 regions for each month of the year, where each panel refers to a region. The data is converted to local time zone.

## 5 Discussion

### 5.1 Uncertainty prediction

Despite the use of optimal interpolation to merge in-situ measurements and model outputs, the STEMMUS-SCOPE simulations still exhibit several limitations and biases. Overall, STEMMUS-SCOPE exhibited the best performance in Rn while the simulation of LE and H were comparable. The model exhibited better performance in simulating LE for forest sites characterized by relatively high LAI, while H was less accurate for wetlands (WET). Biases in different variables included an underestimation of LE from January to June (except for EBF), an overestimation of H in DBF, ENF, and MF, and an underestimation of H in WET. GPP was consistently underestimated across all vegetation types (Wang et al., 2025). Additionally, the lack of observation depths for G and corresponding soil moisture data made calibration challenging, limiting fair comparisons between observed and simulated G (Tang et al., 2024). Due to the PLUMBER2 dataset's time span (1992–2018), validating simulated SIF against tower-based or high-resolution remote sensing SIF was not feasible. Instead, we examined the correlation between modeled SIF and GPP (both observed and modeled), revealing a strong proportionality between them. These results suggest that while STEMMUS-SCOPE performs reliably under certain vegetation conditions, its applicability may be limited in ecosystems with high heterogeneity or lacking comprehensive observational data.

Due to the biases and limitations in STEMMUS-SCOPE simulations, we introduced an uncertainty prediction, where we quantified the uncertainty for each timestep and grid using standard deviation (std) derived from our training and testing samples (see Supplementary S1). Specifically, we calculated two kinds of std with two different approaches: daily std (RF_std1), and std for the same time step (RF_std2). The testing performance is shown in Table 4-5. Generally RF_std2 model has better performance than RF_std1 on three testing sets, because data in the same day has more fluctuations than the data in the same timestep over the same hemisphere and same IGBP type.

We also provide a time series example in one testing_space station: AU-GWW (Figure S4-S5), demonstrating that our FluxHourly not only predicted fluxes, but also computed the uncertainty of the predicted fluxes. The shaded region in Figure S4-S5 represents the prediction uncertainty ($\pm 1$ std). A wider shaded area indicates higher uncertainty, suggesting less confidence in the model's prediction for that time period. Larger uncertainty bands, particularly around peak fluxes, suggest greater variability in model performance in periods with higher flux values.

Table 4 Performance metrics obtained by RF_std1 on three testing sets

| Target variables | Testing_random | | Testing_time | | Testing_space | |
|---|---|---|---|---|---|---|
| | RMSE | r score | RMSE | r score | RMSE | r score |
| Rn (W m$^{-2}$) | 12.42 | 0.79 | 15.77 | 0.63 | 18.01 | 0.48 |
| LE (W m$^{-2}$) | 4.24 | 0.91 | 5.75 | 0.82 | 5.95 | 0.68 |
| H (W m$^{-2}$) | 5.74 | 0.84 | 8.14 | 0.67 | 10.89 | 0.43 |
| G (W m$^{-2}$) | 3.73 | 0.86 | 4.73 | 0.81 | 5.39 | 0.47 |

| Target variables | | | | | | |
|---|---|---|---|---|---|---|
| GPP (umol m$^{-2}$ s$^{-1}$) | 0.31 | 0.94 | 0.42 | 0.87 | 0.47 | 0.78 |
| SIF685 (W m$^{-2}$ um$^{-1}$ sr$^{-1}$) | 0.006 | 0.87 | 0.009 | 0.75 | 0.009 | 0.59 |
| SIF740 (W m$^{-2}$ um$^{-1}$ sr$^{-1}$) | 0.03 | 0.88 | 0.04 | 0.77 | 0.05 | 0.65 |

Table 5 Performance metrics obtained by RF_std2 on three testing sets

| Target variables | Testing_random | | Testing_time | | Testing_space | |
|---|---|---|---|---|---|---|
| | **RMSE** | **r score** | **RMSE** | **r score** | **RMSE** | **r score** |
| Rn (W m$^{-2}$) | 9.18 | 0.92 | 9.79 | 0.91 | 11.87 | 0.88 |
| LE (W m$^{-2}$) | 3.46 | 0.92 | 3.90 | 0.90 | 5.88 | 0.79 |
| H (W m$^{-2}$) | 4.65 | 0.91 | 4.93 | 0.90 | 6.01 | 0.88 |
| G (W m$^{-2}$) | 3.05 | 0.90 | 3.22 | 0.88 | 4.57 | 0.70 |
| GPP (umol m$^{-2}$ s$^{-1}$) | 0.29 | 0.93 | 0.33 | 0.91 | 0.47 | 0.77 |
| SIF685 (W m$^{-2}$ um$^{-1}$ sr$^{-1}$) | 0.005 | 0.93 | 0.005 | 0.92 | 0.007 | 0.87 |
| SIF740 (W m$^{-2}$ um$^{-1}$ sr$^{-1}$) | 0.02 | 0.93 | 0.03 | 0.92 | 0.03 | 0.87 |

## 5.2 Uncertainty from scale mismatch

There is a scale mismatch between the gridded meteorological data and in-situ forcing data. Using gridded meteorological data in the upscaling introduces additional uncertainties in generating global flux products because the RF model was trained on site level (Zeng et al., 2020). The additional uncertainty due to gridded meteorological data is assessed using several different meteorological forcing products (Figure 9), including in-situ data, ERA5-Land (FluxHourly used it as input), and CERES (FLUXCOM used CERES for Rin as input)). CERES has a spatial and temporal resolution of 1 degree and hourly,

respectively.

Scale mismatch exists in every predictor variable, in which Rin is the most important predictor variable. Therefore, Rin was compared among different gridded datasets (ERA5-Land, CERES) and in-situ Rin (Figure 9). This experiment was carried out at 57 stations which have data in 2014. We plot the probability density function (PDF) of difference between ERA5-Land and in-situ Rin (blue), and difference between CERES and in-situ Rin (red). The PDF of errors indicates that both

ERA5-Land (blue) and CERES (red) exhibit error distributions centered around zero, suggesting that their deviations from observed data are generally small. However, the ERA5-Land distribution is more peaked, with a higher density near zero, indicating lower variability and more stable errors. In contrast, the CERES error distribution is broader and flatter, suggesting a wider range of deviations. Additionally, both distributions exhibit long tails, implying the presence of occasional large errors, albeit with low probability. This means the scale mismatch in predictor variables could lead to

underestimation of fluxes. Specifically, coarse-resolution gridded data tend to smooth spatial heterogeneity in surface radiation. When local high-radiation areas are averaged with surrounding low-value regions, the resulting grid-mean Rin

becomes smaller than the true value at flux tower locations. Because Rin is the dominant driver of energy partitioning, this underestimation propagates through the energy balance and results in lower simulated LE. In other words, grid-scale aggregation dampens local extremes and introduces bias propagation from Rin to surface fluxes, explaining part of the systematic underestimation observed in our results.

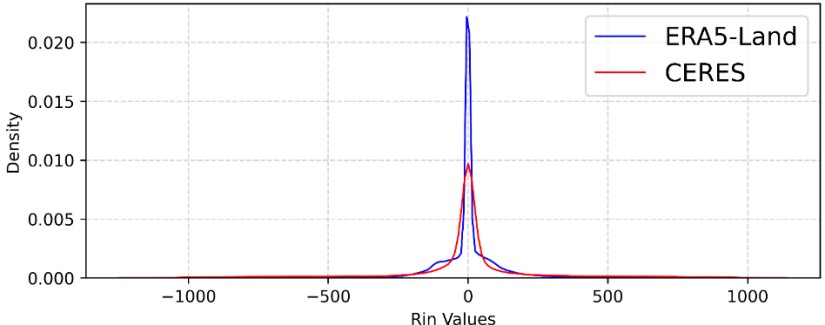

Figure 9. Rin from in-situ, ERA5-Land, and CERES in 2014

**6 Conclusion**

This study used STEMMUS-SCOPE output and in-situ measurement data to develop a RF algorithm based emulator and generated a global fluxes dataset. First, STEMMUS-SCOPE was deployed to simulate land surface fluxes over 170 PLUMBER2 sites. Despite its ability to simulate consistent fluxes at any selected location and supplement variables that are missing or not available in in-situ measurements, it is important to acknowledge discrepancies between STEMMUS-SCOPE simulations and in-situ measurements. We then used optimal interpolation to merge STEMMUS-SCOPE simulations and PLUMBER2 measurements to reduce such discrepancies and to maintain maximal consistency. The optimally interpolated results were then used as training and testing data-pairs to develop the STEMMUS-SCOPE emulator RF_OI using multivariate RF regression algorithm. Seven target variables were predicted simultaneously on global scale at a spatial and temporal resolution of 9 km and hourly respectively, including net radiation (Rn), latent heat flux (LE), sensible heat flux (H), soil heat flux (G), gross primary productivity (GPP), as well as sun-induced fluorescence at 685 nm and 740 nm (SIF685, SIF740). The results show that the generated global land surface fluxes have comparable or better accuracy with existing data products, when evaluated at their respective spatial and temporal resolution, but have much better spatial and temporal features. Additionally the emulator also provides estimates of uncertainty for each variable. We conclude that this new global fluxes dataset, named FluxHourly, can serve as a valuable resource for studying ecosystem responses to climate extremes on global scale.

## Data availability and code availability

FluxHourly is available on https://doi.org/10.11888/Terre.tpdc.302319 (Han et al., 2025a).

The FluxHourly dataset is organized into separate files for each month, with each file being approximately 40 GB in size. The dataset is referenced to the WGS84 coordinate system and includes seven output variables: Rn_OI (net radiation, Rn), LE_OI (latent heat flux, LE), H_OI (sensible heat flux, H), updated_Gtot (ground heat flux, G), Actot (gross primary productivity, GPP), SIF685, and SIF740.

Code is available in https://doi.org/10.5281/zenodo.17697920 (Han et al., 2025b).

## Author contributions

YZ, BS, FA, QH conceptualized and designed this study. QH, YW, FN, YL wrote the codes. QH did the analysis and wrote the original draft. YZ, BS, YW, FA, FN, YL provided guidance and technical inputs to this study. All authors participated in the discussions and provided guidance and advice throughout the experimental design and all reviewed the manuscript. All authors have read and agreed to the published version of the manuscript.

**Competing interests:** The authors declare that they have no conflict of interest.

## Acknowledgements

The research presented in this paper was funded in part by the China Scholarship Council (grant no.202004910427). We are grateful to the SURF for the use of the high performance computing server Snellius which allows us to perform our study efficiently and for their technical help. The authors would like to thank the eScience center colleagues for their technical help with the great cooperation in the EcoExtreML project. This work used the Dutch national e-infrastructure with the support of the SURF Cooperative using grant no. EINF-6614 and EINF-12364. This research has been funded by The Netherlands Organisation for Scientific Research (NWO) KIC, WUNDER project (grant no. KICH1. LWV02.20.004), Netherlands eScience Center, EcoExtreML project (grant ID. 525 27020G07) and Water JPI project "iAqueduct" (Project number: ENWWW.2018.5). In addition, this study was supported in part by the ESA ELBARA-II/III Loan Agreement EOP-SM/2895/TC-tc and the ESA MOST Dragon V and VI Program.

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
