# Peer review of "FluxHourly: Global long-term hourly 9 km terrestrial water-energy-carbon fluxes"

_Earth System Science Data, 2025_

## Author Comment (AC1)

We sincerely thank the Editor, Associate Editor and Reviewers for handling and taking the time to read and review our manuscript. We greatly appreciate the reviewer's insightful and constructive remarks. They have helped us improve both the scientific rigor and the readability of the manuscript. Below we address each comment in turn. Line numbers refer to the clean revised version unless otherwise indicated.

The diurnal cycles of terrestrial ecosystem gas excanges determine the land-atmosphere interaction and land ecosystem function feedback to climate change. This study first incorporated 170 flux in-situ observations covering 11 ecosystem types to train and test the canopy irradiative tansfer model (SCOPE), Then latent heat flux, sensible heat flux, soil heat flux, GPP and SIF for each flux site generated by the SCOPE model. Finally,radom forest model machine learning method integrated with global gridded meteorology and remote sensing was applied to interpolate the site-level variables to global gridded hourly fluxes. This is an interesting study, and fill the scope of ESSD. And are attractive to the community.

However, I have some major concerns for its current version.

Comment 1: For the introduction, the author did not refer to the STEMMUS-SCOPE. I guess the authors try to use the SCOPE model to retrive SIF timeseries for each flux tower? Then I suggest to illustrate the significance of SIF in generating the global hourly gridded fluxes, since the authors have refered SCOPE model in the Abstract.

Response 1: We thank the reviewer for pointing out the need for clarification. In the revised introduction (lines 68-74), we have explicitly introduced STEMMUS-SCOPE model and clarified its role in our workflow. In our study, SIF is treated as an output variable rather than an input driver. Specifically, STEMMUS-SCOPE simulates hourly SIF alongside the five other flux variables at 170 PLUMBER2 sites. We then used simulated SIF time series as a target in our random forest upscaling. From this, the RF model generates a global 0.1° hourly SIF product at 685 and 740 nm (along with the other five variables) based on gridded remote-sensing and meteorological predictors.

The significance of SIF is briefly illustrated in lines 39-41, and we have added more sentences to illustrate its significance in lines 41-43. Including SIF explicitly as an output can reflect diurnal and stress-driven photosynthetic dynamics, thereby improving the physiological interpretability of our global hourly flux dataset.

For the result, site-level training and test.

Comment 2: 1.The author used site-level GPP and SIF to drive the RF interpplation(RF_OI)? Please show the accuracy comparison between GPP_scope and GPP_EC .

Response 2: Thank you for raising this important point. The comparison for GPP_STEMMUS_SCOPE and GPP_EC was already done in our another paper (Wang et al. 2025). https://www.nature.com/articles/s41597-025-05386-x

This is the text from the above mentioned paper: "The simulation of GPP is slightly more accurate than that of NEE. For GPP, the KGE ranged from -0.35 to 0.93 with a median value of 0.55 (median $R^2$, RMSE, rRMSE, and rSD values are 0.64, 3.79 μmol m$^{-2}$ s$^{-1}$, 6.15%, and 0.18, respectively)."

For SIF, we only used output from STEMMUS-SCOPE (due to the lack of in-situ data, but we compared our product with TROPOMISIF in Figure S9).

**Comment 3:** 2. Figure 3 and Figure 4 are good ways to show the technical issue of RF_OI. But I suggest to add analyze the diurnal variatons of GPP/LE/H for each IGBP class directly between the SCOPE output and EC tower. For example the the comparison (SCOPE v.s. EC tower) of mean diurnal cycle within one year for each IGBP class? within one season? This could tell the readers of message from IGBP classes.

Response 3: Thanks for pointing this out. We have made a new figure to compare GPP/LE/H for each IGBP class between STEMMUS-SCOPE output and EC tower observations. For each IGBP class, we first check which year most of the stations have data, then this year was chosen to make plot. For each IGBP class, we labelled the year we chose for plotting, and the number of stations in that year, and the total number of stations for each IGBP. This new figure is Figure S13 in Supplementary S5 (Section 5). The old S5 was changed to "S6".

**Closed shrublands | 2005**
Number of stations in 2005=2  /  total number of stations in this igbp=2

[Figure]

[Figure]

[Figure]

In-situ —— STEMMUS-SCOPE ——

**Croplands | 2005**
Number of stations in 2005=14  /  total number of stations in this igbp=17

[Figure]

[Figure]

[Figure]

In-situ —— STEMMUS-SCOPE ——

**Deciduous Broadleaf Forests | 2004**
Number of stations in 2004=12  /  total number of stations in this igbp=20

[Figure]

[Figure]

[Figure]

In-situ —— STEMMUS-SCOPE ——

**Evergreen Broadleaf Forest | 2003**
Number of stations in 2003=8  /  total number of stations in this igbp=16

[Figure]

[Figure]

[Figure]

In-situ —— STEMMUS-SCOPE ——

**Evergreen Needleleaf Forests | 2004**
Number of stations in 2004=22  /  total number of stations in this igbp=36

[Figure]

[Figure]

[Figure]

In-situ —— STEMMUS-SCOPE ——

**Grasslands | 2005**
Number of stations in 2005=19  /  total number of stations in this igbp=40

[Figure]

[Figure]

[Figure]

In-situ —— STEMMUS-SCOPE ——

[Figure]

For the global gridded fluxes,

Comment 4: 1. If I do not miss the global patterns of hourly products, I did not find the global mean (magnitude) and trends map for each three wate-carbon-energy flux.

Response 4: This is a good point and we agree that the global mean and trends map are important. We have calculated 2001-2020 mean (because 2000 miss data for Jan and Feb, so it is excluded) and trend maps for 2001-2020 annual mean. This new figure is Figure S14 in Supplementary S5 (Section 5).

[Figure]

[Figure]

Comment 5: 2. And then inter-comparison of the mapping pattern between RF_OI and existing hourly products such as FLUXCOM. Currently, the author only show the intercomparison map of LE (Figure 6)?

Response 5: Thanks for noticing this. Figure S9 shows the inter-comparison of Rn, H, GPP and SIF740. In lines 249-276, we described that "Figures 6 and S9 illustrates the spatial distribution of Rn, LE, H, GPP and SIF740 across different datasets." with detailed descriptions.

In line 251, we modified from "Figures 7 and S8" to "Figure 6 and S9". This was an error.

In line 272, we modified from "Figure S8" to "Figure S9". This was an error.

Comment 6: 2. Please explain the rational for the slected 8 regions to decompose the global product. Why not select the global plant function type classes or K-G climate classes.

Response 6: Actually the selected 8 regions were based on K-G climate classes, in lines 283-286. These 8 regions cover five main climate zones, and the K-G climate classes each region covers is listed in Table 3.

Comment 7: 3.Also, the author showed the diurnal cycle of global LE and GPP hourly fluxes for the 8 regions, respectively, but we did not find the 8 regions analyze for global hourly fluxes.

Response 7: It is described in lines 289-306, not only hourly LE and GPP, but also Rn, H, and G which are corresponding to Figures S10-S12.

**Reference**

Wang, Y., Zeng, Y., Alidoost, F., Schilperoort, B., Song, Z., Yu, D., Tang, E., Han, Q., Liu, Z., & Peng, X. A physically consistent dataset of water-energy-carbon fluxes across the Soil-Plant-Atmosphere Continuum. *Sci. Data, 12*, 1146, 2025

---

## Author Comment (AC2)

We sincerely thank the Editor, Associate Editor and Reviewers for handling and taking the time to read and review our manuscript. We greatly appreciate the reviewer's insightful and constructive remarks. They have helped us improve both the scientific rigor and the readability of the manuscript. Below we address each comment in turn. Line numbers refer to the clean revised version unless otherwise indicated.

Land surface water-energy-carbon fluxes are key for understanding Earth's climate system. However, high-resolution data on water-energy-carbon fluxes at finer temporal scales remain limited. This study produced a new data that estimates these exchanges hourly during 2000-2020 by using STEMMUS-SCOPE model, field measurements, and machine learning with satellite and meteorological data. I believe this dataset could provide valuable insights into diurnal variability and finer-scale land-air processes. I recommend that this paper be accepted for publication after addressing the following comments.

Comment 1: 1. Note that estimating the water-energy-carbon fluxes at regional to global scales depend on interpolation processes. The authors applied the optimal interpolation to merge Rn, LE, and H from STEMMUS-SCOPE simulations with eddy covariance observations. Can the authors provide a reason or explanation to why this interpolation is reasonable or why this method can reduce the interpolation errors in the best possible way?

Response 1: Thank you for this insightful comment. We agree that fusing in-situ measurements with model simulations requires a principled and robust approach. In this study, Optimal Interpolation is a variance based data assimilation approach, to combine Rn, LE, and H from STEMMUS-SCOPE simulations and eddy covariance observations at flux tower sites. Optimal Interpolation assumes that both model outputs and observations contain errors characterized by their error variances and covariances. The method provides the best linear unbiased estimate (BLUE) of the target variables by minimizing the expected mean-square error, which is mathematically equivalent to a weighted least squares solution::

$$x = \frac{var_{model}}{var_{obs} + var_{model}} * x_{obs} + \frac{var_{obs}}{var_{obs} + var_{model}} * x_{model}$$

In this formulation, daily variances for each data source were calculated based on 48 half-hourly values. The idea is to assign greater weight to the source that shows less fluctuation within a day, reflecting more stable data quality. This approach is especially beneficial in handling periods where eddy covariance observations may be less reliable. For instance, in situations such as nighttime periods or rainfall events where eddy covariance data tend to be noisy or unreliable, the STEMMUS-SCOPE simulations which is physically constrained receives more weight.

We have revised the description of section 3.4 Optimal Interpolation with the above explanation (Line 151-161). We observed that this simple and computationally efficient scheme can preserve diurnal variability while smoothing extreme outliers. The fused fluxes generated via optimal interpolation were exclusively at flux tower locations to enhance the quality of the training data for Random Forest model. The Random Forest model, driven by satellite and meteorological inputs, was then used to generate the global gridded fluxes.

Comment 2: 2. Section 3.5: The authors said that they used three commonly used statistical evaluation metrics. What's the third one, except for RMSE and r?

Response 2: Thanks for noticing this. This was a typo, should be two instead of three. We have changed "three" to "two" in line 164.

Comment 3: 3. The authors should acknowledge the limitations and biases in the STEMMUS-SCOPE simulations in the Discussion Section.

Response 3: We thank the reviewer for this valuable comment. In fact, we have acknowledged and discussed the limitations and biases of the STEMMUS-SCOPE simulations in the Discussion Section 5.1 (lines 319-328). We elaborated on the model performance and its varying accuracy across variables and vegetation types. We have now revised the paragraph slightly to more explicitly frame it as a discussion of model limitations and biases to improve clarity (lines 317-319).

In addition, we have now added a concluding sentence in lines 328-329 to this paragraph to explicitly emphasize the limitations in model applicability: "These results suggest that while STEMMUS-SCOPE performs reliably under certain vegetation conditions, its applicability may be limited in ecosystems with high heterogeneity or lacking comprehensive observational data."

Specific comments:

Comment 4: 1. line 12: "First the integrated STEMMUS-SCOPE model" ---> "First, …" Suggested to separate by a comma. Similarly, line 15 ---> "Next, …"

Response 4: Thanks for your suggestion. We have added comma in line 12 and 15.

Comment 5: 2. line 124: What does "Method ML" mean?

Response 5: Thanks for pointing this out. We have changed "Method ML" to "Machine Learning Method".

---

## Author Response (AR2)

**Report 1**

We appreciate the reviewer's positive evaluation of our manuscript.

**Report 2**

Surface water, energy, and carbon fluxes are critical to the study of the Earth's climate system, yet high-temporal-resolution global datasets remain relatively scarce. This study utilizes hourly-scale data from 2000 to 2020, combining the STEMMUS-SCOPE model with PLUMBER2 site observations, satellite remote sensing, and gridded meteorological data. Machine learning methods are employed to generate the global FluxHourly dataset at a 9 km resolution. The research integrates multiple site observation data with the SCOPE model, addresses data instability at sites through optimal interpolation, and extends the data globally using random forests. This work fills the gap in high-resolution flux data and provides an important tool for studying land-atmosphere interactions and ecosystem responses. The methodology is innovative, and the scientific value is significant. It is recommended for minor revisions with addressing the following issues.

1. OI appears to enhance site-level data quality by integrating STEMMUS-SCOPE simulations and PLUMBER2 observations. However, specific comparisons of results before and after OI processing are not provided, which would help better understand its actual effectiveness. Additionally, OI is only applied at the site level, while the global product relies on RF extrapolation. This may lead to error propagation or bias adaptation due to differences in data scales. Improvements in site-level data quality may not fully reflect at the global scale. It is recommended to supplement the analysis with comparative results at both site and global levels before and after OI to further validate the effectiveness and applicability of the OI method.

Reply: We thank the reviewer for this constructive suggestion. The main goal of applying OI in this study was to improve the physical consistency and continuity of site-level measurements, including latent heat (LE), sensible heat (H), and net radiation (Rn), by filtering out unrealistic values and short-term spikes. For example, OI helps remove abnormal LE during nighttime and precipitation periods while preserving the general temporal dynamics.

To illustrate this, we have added visual comparisons of time series before and after OI (see new Figure S2). In-situ flux measurements (black) provide the most direct observations but often contain short-term noise and unrealistic fluctuations, particularly during precipitation. The STEMMUS-SCOPE simulations (orange) are smoother and physically consistent but tend to exhibit systematic biases in magnitude. The OI result (blue) dynamically integrates these two sources. As a result, OI follows the observed diurnal dynamics where observations are reliable (e.g., station names in the below figure), while aligning more closely with the model during periods of degraded data quality (e.g., station names in the below figure). In the 6 example sites, OI reduces random variability, avoids unrealistic peaks, and enhances the physical consistency of Rn, LE, and H. This confirms that OI effectively reduces noise and enhances the reliability of site-level inputs used for RF training.

Regarding the reviewer's concern about potential scale differences, we acknowledge that the global product is derived through Random Forest (RF) extrapolation, while OI was applied only at the site level. The performance of the RF extrapolation was evaluated through spatial test, confirming that the model can capture large-scale spatial patterns. Nevertheless, we recognize that further improvement is possible. Future work will focus on enhancing the extrapolation performance by integrating advanced deep learning architectures that can better model nonlinear spatial–temporal dependencies.

[Figure]

[Figure]

2. Feature importance analysis indicates that incident shortwave radiation is the most critical predictor variable. Why is shortwave radiation so important, and is it related to specific physical mechanisms? Further elaboration on the underlying reasons is recommended. Additionally, inherent relationships exist among variables (e.g., between SIF and GPP), which could potentially improve the accuracy of certain outputs. However, the article does not delve into whether these relationships were considered to optimize inputs. It is suggested that the authors explore this potential to enhance data accuracy.

Reply: We thank the reviewer for this valuable comment. The high importance of incident shortwave radiation (Rin) identified by the Random Forest model can be explained by its fundamental physical role in driving land–atmosphere exchanges. This has been added to lines 189-194 in section 4.1.1.

"Rin provides the primary energy input for surface heating, evapotranspiration, and photosynthesis, thereby exerting strong control over both energy and carbon fluxes. For the energy components (Rn, H, LE, G), Rin determines the available energy that can be partitioned into different fluxes (Peng et al. 2021). For the carbon-related processes (GPP and SIF), Rin governs photosynthetically active radiation (PAR), which supplies the energy for carbon assimilation (Nemani et al. 2003; Running et al. 2004). These findings are therefore physically consistent with the observed dominance of Rin in the feature importance analysis."

We thank reviewer's comments on the inherent relationships between SIF and GPP. It is to note that SIF and GPP are outputs of our STEMMUS-SCOPE model emulator. The mechanistic link between SIF and GPP has been described in the STEMMUS-SCOPE model (Wang et al. 2021), and therefore implicitly considered in our emulator.

3. In Fig. 1, the longitude label in the upper right should be 180° instead of 90°. It is recommended to add statistics on the number of sites in each climate zone and mark the spatial locations of the 34 testing_space sites. If these sites are uniformly distributed globally, it would further enhance the persuasiveness of the validation. Additionally, although the selection of the eight typical regions is explained later, it is suggested to clarify this in the title or main text of Fig. 1 to improve reader clarity.

Reply: Thanks for your helpful suggestions. We have corrected the longitude label from 90° to 180° in Figure 1. In addition, we added statistics on the number of sites in each climate zone. In total, the 170 sites cover 17 climate zones, and this information has been added in lines 79-81 in section 2.1.

"The 170 sites cover 17 Köppen climate zones, with the following distribution: Af (2), Am (3), Aw (6), BSh (8), BSk (10), BWk (1), Cfa (22), Cfb (42), Csa (20), Csb (6), Dfa (5), Dfb (13), Dfc (23), Dwa (1), Dwb (2), DWc (1), and ET (5). "

We also marked the spatial locations of the 34 testing_space sites in Figure 1, which were selected based on their IGBP types. Furthermore, the selection of the eight typical regions has been clarified in the caption of Figure 1 (line 85) to improve readability and clarity.

[Figure]

4. The simulated r-values are high (multiple 0.99 in Table 2), but r-values alone are insufficient to convince readers. It is recommended to use scatter plots to display the match between predicted values and actual observations for more intuitive validation of the results.

Reply: Thank you for your constructive suggestion. We have added scatter plots comparing the predicted and observed values for all flux variables to provide a more intuitive validation of the model performance (see new Figure S16) in the Supplementary Material. These plots clearly illustrate the consistency between the predicted and observed fluxes, complementing the statistical results shown in Table 2.

[Figure]

5. Lines 335–336 discuss uncertainty assessment but do not explicitly reference Fig. 9. It is recommended to directly cite the figure to improve clarity. Furthermore, Fig. 9 mentions that "the scale mismatch in predictor variables could lead to underestimation of fluxes." If the purpose is to explain the underestimation of fluxes such as LE, it is suggested to add a dedicated paragraph analyzing the potential mechanisms behind the underestimation. For example, grid-scale mismatches may lead to bias propagation in Rin, thereby affecting fluxes like LE. Supplementing such analyses would deepen the discussion and enhance the credibility of the results.

Reply: I have referred to Figure 9 in line 349. We appreciate the reviewer's insightful suggestion. Following the comment, we have added a paragraph in the discussion (lines 361-366). The new text reads as follows:

"Specifically, coarse‑resolution gridded data tend to smooth spatial heterogeneity in surface radiation. When local high‑radiation areas are averaged with surrounding low‑value regions, the resulting grid‑mean Rin becomes smaller than the true value at flux tower locations. Because Rin is the dominant driver of energy partitioning, this underestimation propagates through the energy balance and results in lower simulated LE. In other words, grid‑scale aggregation dampens local extremes and introduces bias propagation from Rin to surface fluxes, explaining part of the systematic underestimation observed in our results."

6. The supplementary materials contain a considerable amount of disorganized information. It is recommended to reorganize them into a clear structure to improve readability.

Reply: We have reorganized the supplementary materials. The pages number became 25 from 47.

**Report 3**

I do not have further comments, except two very minor ones:

1.It would be great to show years for each site covered and the number of site-year, or site-month or site-hour observations;

Reply: We appreciate the reviewer's suggestion. The temporal coverage of each flux tower site has been comprehensively summarized in our previous publication (Wang et al. 2025), where Figure 2 illustrates the years available for all sites (see below). To avoid redundancy, we have added a citation to that figure in the current manuscript (Section 2.1, lines 77-79).

[Figure]

2.Further reasons why including evergreen broadleaf forests (IT-Lav), mixed forests (BE-Vie), and permanent wetlands (US-Los) for validation should be further highlighted. After revise the minor issues, the manuscript can be accepted for publication.

Reply: We appreciate the reviewer's suggestion. The three validation sites (IT-Lav, BE-Vie, and US-Los) were intentionally selected to represent distinct ecosystem types and climatic conditions. IT-Lav represents evergreen broadleaf forests with high productivity and strong canopy–atmosphere coupling; BE-Vie represents mixed forests with transitional canopy structures and variable moisture regimes; and US-Los represents permanent wetlands with high water availability and dominant latent heat fluxes. These contrasting sites thus encompass a broad range of vegetation structures and energy–water exchange characteristics, providing a robust validation of our approach across different ecosystems. This rationale has been added to the revised manuscript (Section 4.1.2, lines 231–233).

Nemani, R.R., Keeling, C.D., Hashimoto, H., Jolly, W.M., Piper, S.C., Tucker, C.J., Myneni, R.B., & Running, S.W. Climate-driven increases in global terrestrial net primary production from 1982 to 1999. *Science, 300*, 1560-1563, 2003

Peng, L., Wei, Z., Zeng, Z., Lin, P., Wood, E.F., & Sheffield, J. Reducing solar radiation forcing uncertainty and its impact on surface energy and water fluxes. *J. Hydrometeorol., 22*, 813-829, 2021

Running, S.W., Nemani, R.R., Heinsch, F.A., Zhao, M., Reeves, M., & Hashimoto, H. A continuous satellite-derived measure of global terrestrial primary production. *Bioscience, 54*, 547-560, 2004

Wang, Y., Zeng, Y., Alidoost, F., Schilperoort, B., Song, Z., Yu, D., Tang, E., Han, Q., Liu, Z., & Peng, X. A physically consistent dataset of water-energy-carbon fluxes across the Soil-Plant-Atmosphere Continuum. *Sci. Data, 12*, 1146, 2025

Wang, Y., Zeng, Y., Yu, L., Yang, P., Van der Tol, C., Yu, Q., Lü, X., Cai, H., & Su, Z. Integrated modeling of canopy photosynthesis, fluorescence, and the transfer of energy, mass, and momentum in the soil–plant–atmosphere continuum (STEMMUS–SCOPE v1. 0.0). *Geoscientific Model Development, 14*, 1379-1407, 2021